# Psychosocial Risk, Work-Related Stress, and Job Satisfaction among Domestic Waste Collectors in the Ho Municipality of Ghana: A Phenomenological Study

**DOI:** 10.3390/ijerph17082903

**Published:** 2020-04-22

**Authors:** Samuel Yaw Lissah, Martin Amogre Ayanore, John Krugu, Robert A. C. Ruiter

**Affiliations:** 1Department of Work and Social Psychology, Faculty of Psychology and Neuroscience, Maastricht University, P.O. Box 616, 6200 MD Maastricht, The Netherlands; r.ruiter@maastrichtuniversity.nl; 2Department of Agro-Enterprise Development, Faculty of Applied Sciences and Technology, Ho Technical University, P.O. Box HP 217 Ho, Volta Region, Ghana; 3School of Public Health, University of Health and Allied Sciences, PMB 31 Ho, Volta Region, Ghana; mayanore@uhas.edu.gh; 4KIT Royal Tropical Institute, P.O. Box 95001, 1090 HA Amsterdam, The Netherlands; J.Krugu@kit.nl

**Keywords:** psychosocial risk, work-related stress, job satisfaction, solid waste, domestic waste collectors, safety standard, Ghana

## Abstract

Domestic waste collectors play key roles in the collection and disposal of solid waste in Ghana. The work environment and conditions under which domestic waste collectors operate influence their job satisfaction ratings and health outcomes. This study investigated psychosocial risk factors, work-related stress and job satisfaction needs among municipal solid waste collectors in the Ho Municipality of Ghana. A phenomenological design was applied to collect data among 64 domestic waste collectors, 12 managers, and 23 supervisors of two waste companies in Ho Municipality, Ghana. Data were collected from June–August 2018 using in-depth interview and focus group discussion guides. Interviews were supplemented by field observations. Data were analyzed using inductive and deductive content procedures to form themes based on the study aim. Four themes emerged from the study. The study results revealed that domestic waste collector’s poor attitudes and safety behaviors such as not wearing personal protective equipment, poor enforcement of safety standards by supervisors and managers, and work-related stress caused by poor working environments impact negatively on domestic waste collector’s health and safety. Other factors such as poor enforcement of standard company regulations, poor work relations, non-clear work roles, lack of social protection to meet medical needs, poor remuneration, negative community perceptions of domestic waste collectors job, work environments, and workloads of domestic waste collectors were reported to negatively impact on work stress and job satisfaction needs. In conclusion, the findings are important in informing the necessary waste management policies aimed at improving decent work environments, as well as improving the health and well-being of domestic waste collectors in both the formal and informal sectors in Ghana.

## 1. Introduction

Globally, effective management of solid waste is both an economic and human security concern. In recent years, the problems associated with the management of solid waste have become acute [1,2] in developing countries because of rapid urbanization and poor spatial planning in cities [3,4]. For developing countries to meet the 2030 target for clean and healthier cities [SDG 11], the active participation of governments, private sector players, and the support of the citizenry is necessary [2,3,5]. The active participation of relevant stakeholders in solid waste management ensures synergy and continuity in dealing with the technical, environmental, financial and socio-cultural exigencies related to waste management [5,6,7].

In Ghana, a Waste Management Department was established in 1985 to manage urban waste [8]. However, poor waste management practices by Metropolitan, Municipal and District Assemblies (MMDAs) led to a policy shifts towards public–private sector partnership (PPP) in managing waste [9]. In the PPP arrangement, an organizational culture was built in which public actors such as MMDAs work with large to medium size private companies who employ Domestic Waste Collectors (DWCs) to collect and dispose of solid waste. Yet still, the PPP arrangements for managing waste in Ghana have been generally described as poor [10]. Evidence in Ghana suggests poor organizational culture attributes such as poor stakeholder engagements and lack of transparency in waste management contracts and poor quality assessment procedures regarding solid waste management in Ghana [10,11,12]. Often neglected in many of these contractual arrangements is improving the welfare of DWCs who are integral to the success of any PPP arrangement in managing waste.

The DWCs play critical roles along the waste management value chain, particularly in developing countries where the collection, sorting, and disposal of solid waste are largely performed manually [13,14]. DWCs daily exposure to waste during collection and disposal makes them vulnerable to occupational health hazards such as skin irritation, bronchitis, hypersensitivity pneumonitis, and dermatitis among others [14,15,16]. Poor measures regarding work design, organizational policies, and management practices contribute to work-related stress for DWCs, which is reflected in both personal and environmental outcomes such as depression and workplace violence [17,18,19,20,21]. Furthermore, persons engaged in the solid waste collection are often people of low socioeconomic and educational status who often are not prioritized regarding their health and safety at the workplace. However, DWCs play an important role in helping to make cities and environments cleaner [22,23,24].

Earlier studies in Ghana on DWCs have reported work-related accidents, absenteeism from work due to varied illness, and deaths among DWCs [9,25,26,27]. Despite DWCs’ important roles in collecting and processing waste in Ghana, research on their physical and psychosocial health and well-being is limited. A recent study found high reports of respiratory infections, reduced pulmonary function and a higher prevalence of dermatological diseases among DWCs in Tema Municipality of Ghana [28]. Respiratory tract symptoms, headaches, body pains, and stomach discomfort were also reported in another study in the city of Accra [29].

To contribute further to the understanding of the occupational needs and hazards among DWCs in the context of solid waste collection and disposal in Ghana, this qualitative study investigated psychosocial risk factors, work-related stress, and job satisfaction needs among DWCs in the Ho Municipality, Ghana. Specifically, the objective of the study was to understand the views and experiences of DWCs and their managers or supervisors regarding personal health and safety risks, work-related stress, and job satisfaction needs in two waste companies in Ghana. The findings are relevant to inform policy-makers of the occupational and health hazards that DWCs encounter daily, and to inform future waste management policies aimed at improving sanitation and hygiene procedures, as well as improving the health and well-being of DWCs in both the formal and informal sectors in Ghana.

## 2. Materials and Methods

### 2.1. Study Design and Ethical Approval

A qualitative phenomenological approach was applied to understand both DWCs’ and their managers and supervisors’ views and experiences regarding risk, work-related stress, and job satisfaction needs along the waste management value chain in the study setting. The phenomenological design was applied since the research team were interested in understanding the perspectives and lived experiences of the phenomenon of interest [30,31].

The Ethical Review Committee Psychology and Neuroscience (ERCPN) of the Faculty of Psychology and Neuroscience, Maastricht University (approved research line ERCPN-188_10_02_2018) and Ghana Health Service Ethics Review Committee (GHSERC 08/05/17) approved the study. During the study, each participant completed a consent form before participating, after being informed about the objective and procedure of the study. Participants’ identities were concealed by ensuring that participants do not mention their names on tape during the recording and interviewing processes. The study participants were assured that the information provided is handled with confidentiality and analysis of the data is done at the aggregate level of the group to ensure anonymity. The research team assured the participants that the information shared with the researchers will not be used to harm their job in the future. This generated a lot of confidence among the study participants and ensured a higher level of participation during the interviews.

### 2.2. Study Setting and Population

The study was conducted in the Ho Municipality, located in the Volta Region of Ghana. The 2018 projected population of Ho Municipality is estimated at 213,960, comprising 105,195 males and 108,765 females [32]. Spatial planning in the municipality is poor [33], with solid waste being poorly managed, and waste collection, management, and disposal largely the responsibility of the household members. The Municipality is regarded as the commercial center of the Volta Region due to its strategic location and its proximity to the Republic of Togo as well as easy crossing to Districts and Regional capitals such as Accra. The Municipality was chosen for the study because of its solid waste management challenges, and the state of environmental conditions in the municipality suggests a lot more needs to be done. Two waste management companies operating in the Municipality were included in the study. The waste management companies and participants were anonymized (the two participating companies are referred to as Company A and B). The two waste companies work with several DWCs to manage solid waste in the Ho municipality. DWCs contracted by the two waste companies as waste collectors, and the waste companies’ managers and supervisors, were eligible participants in this study. Waste company managers are responsible for recruiting and training DWCs, while supervisors are principally involved in monitoring the daily activities of the DWCs.

### 2.3. Study Instrument

In-depth interview (IDI) and focus group discussion (FGD) guides were developed for conducting interviews among managers/supervisors and DWCs, respectively. On average, IDIs lasted 50 min, while FGDs lasted for 1:15 min. The design of the interview guide was informed by the core attributes of the Copenhagen Psychosocial Questionnaire (COPSOQ) [34]. COPSOQ aims to improve and facilitate research and intervention design at workplaces. The study adapted the short, medium and the long versions of the Copenhagen Psychosocial Questionnaire (COPSOQ) scales. All the versions were modified to meet the objective(s) of this study. Specifically, we created a mix of scales to measure psychosocial risk, work-related stress, and job satisfaction. The interviews with DWCs were partly conducted in “Ewe”, the indigenous language in the study area and English for non-Ewi speaking. Interviews with managers and supervisors were conducted in English. The interview guide was pilot tested among ten DWCs, five each from the two companies. No content changes were made to the instrument except for some small textual edits. All field data were collected between June and August 2017.

### 2.4. Sampling and Interview Procedures

The principal investigator (SYL) consulted the Municipal Environmental Health Officers (MEHOs) to identify study sites in the Ho Municipality to conduct the study. A sampling frame was created after obtaining anonymized details of DWCs from the two companies operating in the Municipality. Overall, 70 DWCs met the eligibility criteria of having minimal work experience as DWC of one-year. Different work profiles among the DWCs were included in the study such as cleaners, sweepers, janitors, and drivers who worked for the two waste companies handling solid waste along the value chain, 46 from Company A and 24 in Company B. However, six (6) DWCs were excluded because of their non-availability at the time of data collection. We had 64 DWCs for the interviews. Company A had a total of 43 DWCs listed, while company B had 21 DWCs listed for our sampling frame. We applied proportionate sampling to ensure equal representation of participants in the interviews. Convenience sampling is a non-probabilistic sampling procedure that results in the recruitment of subjects of the population that are easily accessible to the data collectors [35,36]. A total of 25 managers and supervisors from company A, and ten from company B, were included in the study. They were selected based on the inclusion criteria of a minimum of one-year continuous work experience with their present company. 

Three research assistants (RAs) were recruited to assist in data collection based on their level of education, prior demonstrated experience and performance during training in conducting field research observations, and a good understanding of ethical procedures in research. Fluency in the local language (Ewi) and good background knowledge of the study setting were also considered in recruiting RAs. The RAs were trained on how to undertake field observations and to administer informed consent to study participants. The principle investigator led all interviews, while RAs assisted to take short notes during the conduct of these interviews. Short notes from the three RAs were later corroborated with audio recordings after transcriptions were concluded and there were no significant differences between the transcripts and the notes from all RAs. During field data collection, RAs visited dumping sites and documented field observations on types of solid waste generated, methods of solid waste collection and disposal, as well as use of Personal Protective Equipment (PPE) in a notebook.

After a prior introduction and familiarization visits to the two companies, dates were fixed for the principal investigator to meet the management of the two companies to explain the purpose of the study further and to gain their support to conduct the study. In these meetings, the principal investigator and management of the companies agreed on the timeline for data collection among DWCs, supervisors, and managers. A week before the interviews, DWCs were informed through their managers of the schedule for data collection by the team. During the period of data collection, DWCs were requested to select a place and agreed time at their convenience for the interviews.

Six focus group discussions were conducted as follows: four FGDs were conducted in company A, while two FGDs were conducted in company B. Four of the six FGDs had ten participants each participating in the discussions. Two FGDs were used: one, in company A, had 13 participants, while another FGD, in company B, consisted of 11 participants. The use of focus discussion made the participants express their opinions and real-life experiences on research topics. All six FGDs were conducted at separate times and locations. In addition, 35 IDIs were conducted among managers and supervisors. For each IDI respondent, interviews were conducted until at a point where saturation was reached among each interviewee. The principle investigator led all interviews while RAs assisted by taking short notes during the conduct of interviews and field observations. Short notes from the three RAs were later corroborated with audio recordings after transcriptions were concluded and there were no significant differences between the transcripts and the notes from all RAs.

### 2.5. Data Analysis

All interviews were audiotaped and transcribed verbatim to English. Two native Ewe teachers were hired to transcribe Ewe audiotapes into written English and then later translated into English. Each transcriber’s work was cross-validated with the other transcriber. The transcribed MS word documents were exported into QSR NVivo 11.0 software (QSR International, Burlington, MA, USA) for analysis. Data were analyzed using inductive and deductive content procedures. Subjecting the data to inductive analysis, all transcripts were repeatedly read by the research team and relevant statements were extracted from the transcripts. The process was repeated twice to ensure that relevant statements were not left out. All extracted statements were then put into categories to form subthemes. Deductions were further conducted on the subthemes by reading and making deductive inferences, leading to the emergence of four main themes. The final themes were organized according to psychosocial risk factors, work-related stress demands, and job satisfaction needs among DWCs. Relevant quotations from the DWCs, managers, and supervisors, and the deduced meanings, are presented within each theme that emerged.

## 3. Results

### 3.1. Socio-Demographic Characteristics of the Participants.

Table 1 shows the socio-demographic characteristic of the study participants. The mean age of DWCs was 41 years. The majority of DWCs (63%) were married at the time of the study. Participants identified themselves as either Christians (64%), traditionalists (25%) or Muslims (6.3%). Most of DWCs were females (66%). Males constituted 34% of the study population. Almost half of the respondents 29 (45.3%) had no education, 16 (25%) had completed the Middle School Leaving Certificate, and 14 (21.9%) and five (7.8%) had Junior Secondary School or Vocational training, respectively. Besides DWCs, a total of 23 supervisors and 12 managers in the two companies took part in the study. Approximately 82.9% of the managers and supervisors were males and their mean age was 41 years. The average years managers and supervisors had worked in their respective companies ranged from 5 to 15 years.

From the analyses of the transcripts, four main themes emerged: (I) Adherence to and enforcement of safety standards on waste management; (II) Relationships at work and working conditions; (III) Social protection and job insecurity, and (IV) Job satisfaction. These main themes are discussed in detail in the next sections. Table 2 presents further details on how each of the final themes emerged.

#### 3.1.1. Theme I: Adherence to and Enforcement of Safety Standards on Waste Management

A key theme that emerged among study participants was the expression of poor enforcement and adherence to safety standards and practices along the waste collection value chain. DWCs expressed concerns of risky occupational behaviors exhibited by their members such as poor use of PPEs, a practice that exposes them to various health hazards. The narratives among DWCs also show poor enforcement of standard regulation procedures by companies and the state of solid waste collection and disposal. Perceptions of risks among DWCs in relation to waste collection and disposal seems to be quite low and does not make them see the hazards and threats associated with their work. DWCs seem to be aware of the need for protective wear, but often failed to wear them practice, as narrated by one DWC:
“I am aware and most of my colleagues are aware of the benefits of adhering to safety rules and exhibiting positive behaviors on the job, the challenge is keeping positive behaviors such as ensuring that you use PPE or undertake regular hand washing or use of disinfectants for cleaning after your daily cleaning schedules.”*(36-year-old female DWC)*

DWCs’ low-risk perceptions and unsafe occupational practices such as use of PPE were confirmed by the waste company supervisors, who pointed out that most reported company accidents among DWCs were caused by poor adherence to measures regarding personal safety and well-being at the workplace:
“DWC do not pay attention to the potential accidents, diseases, and injuries linked to their job. The issue of solid waste handling and the various steps that need to be taken by the DWCs to ensure their own safety are not taken seriously.”*(Male supervisor—company B)*

##### Adverse Health Outcomes and Work-Related Stress

DWCs are exposed to potential health risks such as physical injuries and self-reported symptoms of respiratory tract infections, irritation, and skin diseases. Some DWCs also avowed frequent cholera attacks, headaches, and body pains, as well as stomach discomfort, as common health conditions frequently experienced by DWCs.
“Hmmm, it is difficult to breathe when smoke and dust enter into my nose and eyes. I feel pains in my chest and coughing as well as eye infections.”*(40-year female DWC)*
“Ehhh you see I woke up early at dawn to walk for about 4 miles from “Titrinu” to Ho to my duty place and start working in the morning by sweeping, collecting, pulling and lifting waste bins into the waste containers for disposal. Sometimes I have pains all over my body especially in my arms, joints and sometimes I have sleepless night.”*(40-year male DWC)*

##### PPE Availability and Use

DWCs’ use of PPE is often influenced by two external factors. First, the companies’ inability to provide PPE to DWCs and secondly, poor enforcement of the use of PPE by the companies’ management. The DWCs that wear little of PPE that is provided are exposed to injuries and diseases such as tetanus, or hepatitis B. As the DWCs would like to do their daily work without the use of PPE, but prefer to handle the solid waste with their bare hands and the few PPE that are provided for the DWCs may be sold to get money for their upkeep. Another issue is that DWCs sometimes set fire to solid waste collected at the disposal sites to burn, since the waste is not sorted and unsafe behavior is practiced by DWCs themselves. One DWC recounted how he undertakes his daily tasks without the needed PPE and equipment (a view shared by the majority of DWCs).


*“The health risk in my workplace is due to the way I go about sweeping and collecting refuse without PPE and using my bare hands to collect the rubbish and other tasks. I do not use the PPE since my PPE are worn-out and have not been replaced after I reported to my supervisor.”*
*(30-year-old male DWC)*

Managers and supervisors responding to the assertion of the company’s not providing PPE acknowledged that there are often delays in receiving new supplies of PPE and other logistical tools, often putting DWCs at risk in the performance of their job. Some DWCs shared their views on poor company arrangements for them when they get injured at work. Their expressions indicate that some of them get no assistance or compensation from their employers for any negligence that may lead to injury or ill health, as recounted by one DWC.


*“What I can say is that there are no procedures in reporting occupational accidents, injuries, and sicknesses in the company that I am aware of. I say this strongly because the company’s management would not respond or attend to me or any of my colleagues in case of injury or accident.”*
*(25-year-old female DWC)*

##### Enforcement of Standard Regulation Procedures

In exploring how company policies and procedures support DWCs’ compliance to safety standards in the performance of their duties, managers in both companies’ agreed that there is a need to strengthen rules and procedures in order to ensure better compliance and accountability in terms of implementing a risk reduction strategies among workers. Most supervisors were of the view that introducing new technologies and reward schemes into tracking waste collection processes can improve DWCs’ compliance with the set rules.


*“We do our best, just that there are so many DWCs that we cannot go-round at the same time. Improving internal monitoring mechanisms and use of new technologies may help to improve compliance levels among DWCs.”*
*(Male supervisor-company B)*


*“When DWCs know they will be rewarded when they comply with safety standards and rules, it will improve the level of compliance, and prevent health-related risk among DWCs.”*
*(Male manager-company A)*

#### 3.1.2. Theme II: Relationships at Work and Working Conditions

##### Work and Interpersonal Relations

DWCs indicated that teamwork and co-operation between them and between their management is required to manage waste effectively. Working along a straight chain of command was reported by DWCs as important in reducing any repetitive job duties and work burnout, as expressed by one DWC focus group discussant.


*“I have worked with this company for about six years. The best way to get things done without much stress is to listen to the command of instructions and work together as a team to get the job done. You cannot do the work alone.”*
*(FGD company A)*

Some DWCs avowed that poor interpersonal relationships were common between them and their managers and explained that such a negative work culture was driven by the company’s working conditions and other work-related structures. Other DWCs were of the view that their companies are not proactive on matters related to their conditions of work, such as benefits and compensations, leading to frequent experience of undue financial stress and burdens on them.


*“Despite the poor nature of our conditions of service, some superior officer’s disregard our efforts at work, exacerbating already poor supervisor-DWC relations and impacting negatively on work conditions.”*
*(45-year-old male DWC)*

##### Non-Clarity on Job Roles and In-Service Training

The narratives of the DWCs suggest that there is lack of clear lines of career progressing in the waste management sector which negatively affects work habits and work stress. On the other hand, the managers and supervisors narrated that continuous on-the-job training for DWCs is intrinsically designed into their company’s policies to reduce worker attrition rates and improve healthy work habits for DWCs. While such training was described as ‘occasional’ by DWCs, some indicated that their experience of the training was useful in helping them manage stress and improve their work-related habits.


*“I was part of a training my company held two years ago on how to manage our stress and improve our work habits. It was beneficial since the lessons have improved my working relations with my colleagues and supervisors. Am not easily stressed up like how I use to be when I first started the job four years ago.”*
*(41-year-old female DWC)*

##### Perceptions of Work Relations among and between DWCs, Managers, and Supervisors

Managers and supervisors of waste companies acknowledged that relationships among DWCs, such as mutual respect for hierarchy, good supervisor–DWC relations, efficient task coordination roles among workers, improvements in incentivized conditions of service, and skill improvement in stress management needs are essential in driving healthy work habits and reducing work-related stress among DWCs.


*“Oh I have a very good working relations with my manager, and this is helping me a lot to improve on my work as a DWC, and I know what my managers expects of me to do on duty.”*
*(48-year-old female DWC)*

#### 3.1.3. Theme III: Social Protection and Job Insecurity

##### Social Protection for Meeting Medical Care Needs

DWCs expressed displeasure that, despite their dedication to improving environmental safety and hygiene in the Municipality, their psychosocial well-being is often ignored by their employers and communities. Their health needs such as health insurance and medical bills were not paid by their employers, so DWCs had to pay out-of-pocket from their monthly wages of Ghc100 (20 USD) to meet their medical bills or pay a fee of Ghc25 for health insurance in order to enable them access health care. Even when they experienced job-related accidents, DWCs explained that they were not guaranteed any medical refunds. Consequently, most of them relied on their families’ limited finances to support their medical bills. One DWC in FGD recounted how he was hospitalized in the line of duty, and had to settle the medical bills after being knocked down by a vehicle, but was barely taken care of by his company, despite his dedicated service to the company over six years. DWCs mentioned ailments such as dog and snake bites, injury by sharp bones, car and bicycle accidents resulting in injuries such as broken legs and hospitalization in the line of work. This was confirmed by the waste managers, who said that record keep is poor with regards to ailments suffered by DWCs and, since the DWCs are casual workers, they have to bear the cost of treatment on their own. This view of “medical neglect” was expressed among many other DWCs.


*“We have no health insurance to meet any medical emergency that may arise from this work. So, while we risk cleaning our cities and town, nobody remembers us by taking care of our health bills when we are ill.”*
*(28-year-old female DWC)*


*“I got knocked down by a moving vehicle after the vehicle ignored roadblocks to slow down. I was hospitalized and had to pay for my own medical bills of Ghc150 with the support of my family before I was discharged from the hospital. My medical bills were not refunded by the company. The only thing the company did was to pay a visit to me at the hospital. I took care of my own hospital bills and treatment.”*
*(35-year-old male DWC)*

Corroborated views by DWC also pointed out that their health, well-being and work environment were often neglected by their employers, although this has a profound influence on their work balance, health, and economic outcomes. The majority of DWCs resorted to self-medication and seeking health care from traditional healers, as they are cheaper even when catastrophic injuries such as a broken leg occur. Nevertheless, they admitted that self-medication and traditional treatments are poorly managed.


*“I will prefer to visit health facility over the traditional healing house. But cost and no insurance is often the reason why I resort to traditional care seeking practice.”*
*(40-year-old female DWC)*


*“While we value health insurance and other physical needs, we think employers must not disregard our emotional health and body needs. They must know that, once we put our lives at risk to clean our cities, they have a role to respect our dignity and meet our psychosocial well-being.”*
*(45-year-old male DWC)*

##### Lack of Proper Contractual Agreement, Poor Remuneration, and Job Insecurity Issues

Managers and supervisors acknowledged that DWCs were employed as ‘casual workers’. Such employment offered weak contractual arrangements and poor adherence to agreements. They were temporal staff, offered an income of Ghc100, sometimes they were not paid for six or more months and whenever they were paid, deductions of Ghc5 were made for their welfare dues, which are not accounted for by waste companies. Participants narrated that such poor working conditions negatively affect their psychosocial and financial wellbeing.


*“The DWC employment conditions and the quality of employment in solid waste management are very poor. In addition, our status as DWCs are very low and we receive very poor remuneration.”*
*(55-year-old male DWC)*


*“You know if I have high level of job satisfaction I would have positive attitudes toward my job as a DWC. But with wages being very low Ghc 100 per month and even irregularly paid for the past seven to eight months. Incentives such as transport, accommodation, and social allowance are non-existent.”*
*(60-year-old male DWC)*

#### 3.1.4. Theme IV: Job Satisfaction

##### Attitude towards Waste Generation and Collection at Household Level

DWCs indicated that the negative attitudes and perception of some community members towards people who are involved in the disposal of solid waste often resulted in uncontrolled stress and poor job satisfaction. DWCs perceived that community members view them as persons of low social status, commanding little respect in society. Supervisors’ views corroborated that of the DWCs assertions of negative social perceptions of their roles in waste management in the city, as recounted by both a DWC and supervisor.


*“But some also insult us like ‘Borla, Borla’ (local parlance for refuse) people and finds it very difficult to accept us in their midst. To them, we are not fit to be amongst them when there is community gathering.”*
*(25-year-old male DWC)*


*“Another issue is the negative perception of the community towards DWCs in solid waste management results in the disrespect for the work and invariably produces low working morals among DWCs and poor quality of their work performance.”*
*(Female manager company B)*

##### Motivation for Being a DWC

DWCs explained that the lack of employment opportunities is the main reason they opted to work as waste collectors. Being hired as a DWC offered some level of income, although this was not a guaranteed flow of income because of funding challenges. While some DWCs expressed dislike for their current job, others reported positive perceptions about their work and their ability to contribute to waste management in the city. Company managers also expressed delight in some DWCs’ efforts and resolve to help manage waste in the municipality despite operating under poor working conditions.


*“I am only just interested in something that I can regularly do which will give me constant income even though I might not be interested in doing that work. My morale is low and depressed with respects to the job content, conditions of service and working environment.”*
*(50-year-old female DWC)*


*“I feel good working since there is no other job instead of staying idle and not doing anything. I could do things which I was not able to do before by contributing to managing waste in the city.”*
*(45-year-old male DWC)*


*“The attitudes of the DWCs have improved toward the good practices of handling waste and they are proud, and love their job. In fact, DWCs have passion for the job, even if their salary is not coming they work.”*
*(Male Manager Company A)*

##### Excessive Workload and Working Environments

DWCs also reported that poor remuneration with a high workload and perceived poor management skills by managers and supervisors frequently affected their job dissatisfaction and increased their vulnerability to several psychosocial risk factors.


*“I work hard to do the tasks given to me although the work is demanding and poor supervision from managers and supervisor. At first, I was glad but because of how things are going now, I am not happy and no award to ‘ginger’ me to work harder. I do not want to be working without my salaries not paid for seven months. As the saying goes ‘An army does not march on empty stomach.”*
*(30-year-old female DWC)*

DWCs recounted that some community members were happy with their valuable service to their communities, despite work-related hazards and poor remuneration issues affecting them as DWCs. DWCs avowed that community perceptions of their job roles and the absence or presence of social support for promoting environmental safety and hygiene have tremendous influence on their job satisfaction ratings as DWC.

“*Some people praise us for the good work we are doing in keeping the environment clean and give us money and food items.*”*(57-year-old female DWC)*


*“When the work environment in which DWC works has a positive influence, it leads to promotion of psychological health, protection of health and safety of DWCs resulting in job satisfaction, confidence and social integration among DWCs. This would go a long way to prevent injuries and diseases.”*
*(Female manager company A)*

## 4. Discussion

This study applied a qualitative design to examine psychosocial factors, work-related stress, and job satisfaction needs among DWCs in the Ho Municipality of Ghana. Although the study did not aim to assess gender differences, we highlight some key gender issues that are worth mentioning. There were fewer females among the study participants at the level of managers and supervisors compared to males. This is the opposite to the amount of DWCs, since more females constituted this level of study participants. DWCs’ job roles are the lowest along the waste management value chain, indicating that females tend to occupy lower job roles in managing waste. This finding is consistent with current waste management operations across Ghana. Some reasons, such as culture, and socio-cultural value systems that regard jobs such as cleaning, sweeping and waste collection as traditional roles for women, may be influencing low socio-economic women’s desire for domestic waste collection job roles, as seen from this study. Policies that will empower women working in the waste management sector to take a leadership position and assume higher responsibility and roles will enhance family livelihoods, decision making, and bridge the inequality in the job role specifications in the waste management sector in Ghana and across similar settings.

The findings suggest that DWCs are frequently exposed to health-related risks and often fail to observe occupational safety practices at the workplace, partly due to not being provided with protective wear such as PPEs. DWCs non-compliance to wearing PPEs, poor risk perceptions towards potential infectious pathogens associated with direct exposure to waste, and belief that dirt is “harmless” can put DWCs at risk of adverse health outcomes. Our findings also show that DWCs work under challenging conditions: their contractual agreement with their employers is often unclear, they are poorly remunerated, hardly receive refresher training on safety issues regarding their work, and do not enjoy healthy working relationships with both colleagues and supervisors. Waste collection is generally perceived by the public as a job for people of low socio-economic standing and DWCs frequently feel stigmatized when people refer to them as “Borla Borla” people.

The finding that DWCs do not often use protective wear to protect them from potential infectious diseases is partly because their managers and supervisors do not pay attention to providing the necessary PPE is worrying. Several studies have linked various adverse health outcomes such as respiratory infections, injuries, skin rashes, lack of concentration at the workplace, anxiety, and loss of sleep to unsafe waste collection practices [14,15,37,38,39]. Despite the health risk associated with their work, one essential health need that is often neglected is meeting medical expenses for DWCs when accessing and using health care services. The inability of some DWCs to afford health care costs often results in self-medication, delays in seeking care or resorting to non-certified herbal practitioners. Asase and colleagues reported that, in Ghana, DWCs often resort to non-certified herbal medicines in times of ill-health, a phenomenon that affects their quality of life in the long-term [40]. The occupational hazards associated with the work of DWCs could be minimized through the consistent use of PPE. However, DWCs recounted working in poor conditions, including frequently not using PPE, and not being adequately trained on safety measures associated with their work to reduce health risks. These conditions seem to inform their poor knowledge in identifying the psychosocial risks associated with their work environment and adopting the necessary health and safety measures. This finding corroborates two Ethiopian studies that reported that poor work environments and lack of adequate training among industry workers on identifying health-related risks and adherence to safety practices contributed to workers’ poor knowledge of occupational health and safety at the workplace [41,42]. These findings do not only call for waste management companies to provide adequate protective wear to domestic waste workers, but also to regularly conduct on-the-job training on the health risks and occupational hazards associated with waste collection.

Our findings also showed that DWCs have low-risk perception towards possible infectious diseases associated with waste collection, which is partly informed by the normative belief that dirt is “harmless”. Such beliefs could mean that even if they are provided with full protective wear, they may still fail to observe safety measures consistently. Although DWCs were of the opinion that they fully understood the implications of dirt having direct effects on health, their expressions and beliefs that not all dirt was harmful was strong enough to prevent their use of PPE, or adherence to other prescribed health and safety rules at their place of work. Additionally, DWCs’ low-risk perceptions from this study were also a result of poor knowledge of the potential health hazards from the direct and indirect effects of the unhygienic handling of solid waste during collection and disposal by DWCs. DWCs’ expressions of low-risk perceptions and belief that dirt was harmless may have been driven by low social esteem, and the low socio-economic and educational levels of DWCs in this study. One repeated assertion by managers and supervisors on DWCs’ normative beliefs and low-risk perception was that these were the result of waste companies’ non-compliance to standard operating procedures on waste management. Workplace policies that provide a system of continuing education and awareness on occupational health and safety can address gaps, misconceptions and poor normative beliefs on the causes of infectious diseases and their prevention among DWCs, as found in this study.

Poor interpersonal relationship among DWCs, their managers, and supervisors can impact negatively on DWCs’ work balance and health outcomes, as narrated by participants of this study. This finding has been corroborated by another study that found poor work and working relations between managers and their subordinates to be associated with subordinates’ reports of mental health conditions such as depression, fatigue, and stress [43]. Specifically related to waste control and management, other studies also found that the lack of mutually respectful relationship experiences of DWCs could be due to the poor nature and conditions of service of DWCs, disregard and poor supervision from their managers, as well as poor communication regarding clarity of the work of waste collectors [43,44]. There is evidence that work-related stress results in a decrease in the quality of relationships with managers, supervisors, and family members [45]. DWCs further indicated that they are more comfortable with managers and supervisors who have their wellbeing at heart. Studies found that the more cordial the relationship between supervisors and managers and workers or DWCs, the more likely it is that the DWCs will perform their duties in a healthy and safe manner [46]. To improve DWCs’ working conditions and build cordial work relations that promote DWCs’ work balance and health outcomes, there is need for workplace strategies to provide safe environments in which healthy work–life balance programs are established and promoted among employers and their employees in order to promote productive work environments.

Furthermore, DWCs revealed that they encounter poor social interactions due to the stigma associated with their work and lack support from supervisors and managers in dealing with the situation, a phenomenon that results in low job satisfaction. Lack of social support for waste collectors impacts negatively on the psychosocial well-being and health outcomes of waste collectors, as previously reported [47,48,49]. DWCs also avowed the existence of poor contractual agreements with their employers, low wage earnings and ‘erratic’ payments of salaries and benefits from employers as contributing to their work-related stress and low motivation, leading to poor job satisfaction. A study [42] conducted in Ethiopia among textile workers corroborates the findings from this study that poor contractual agreements and low wages are psychosocial risk factors that can result in work-related stress and poor job satisfaction among workers. Similarly, psychosocial risks and work-related stress are known to determine job satisfaction in the work environment [50,51].

The findings presented in this study also reveal the stigmatization associated with the job roles of waste collection and disposal. This study found that citizens’ wrongful perception and disregard for the important job role of DWCs in supporting the collection and disposal of waste influences how citizens support Municipal city authorities to manage what waste is generated and disposed of. This wrongful perception by citizens of who “owns” waste or has the responsibility to “take care” of waste often leads to indiscriminate waste disposal, and the poor perception that waste could be disposed of everywhere, since DWCs are available to collect this waste. The stigmatization of DWCs may result in disrespect, isolation, harassment and invariably produces low work ethics and productivity, as recounted by some DWCs during interviews. This study advocates for broader social and legislative policies that ensure that citizens develop a sense of social responsibility in the collective management of waste. Specifically, Municipal authorities will need to develop behavior change communication (BCC) strategies that target improving citizens’ awareness of and support for collective waste management. There is also a need to develop legislative policies that prohibit indiscriminate waste disposal, and ensure support for DWCs’ and waste management companies’ continued roles in managing Municipal waste.

The findings from this study also point to the high expectations and demands from employers for DWCs to meet work schedules given the tons of waste generated in the city on a daily basis, despite the rudimentary manner in which waste is still collected across cities in Ghana. The volume of waste to be collected daily and the rudimentary daily routine of using tricycles and other low-tech equipment for collecting and disposing of waste was identified by DWCs as contributing to work-related stress and poor job satisfaction. Studies across many settings among waste workers corroborate our study finding on workloads and stress-related effects [42,52,53,54,55,56]. There is also evidence to suggest that human resource practices and remuneration or wage policies are important determinants of job satisfaction [57].

To address the concern about the volumes of waste generated and how these could be better managed at the household level, there is need for waste company managers and supervisors to provide ideal equipment and tools to assist DWCs in the effective and appropriate modern standard of solid waste management. This could minimize DWCs’ direct or indirect contact with solid waste to prevent the potential health-related risk of exposure to solid waste among DWCs. For DWCs to be satisfied with their job, there should be a timely and available supply of PPE, a flexible work schedule and reasonable demand of how their work is done to reduce excessive pressure on DWCs or workers, and regular payment of wages to prevent DWCs from using their scanty wages to purchase equipment for the work. There is a need for workplace policies that address flexible work schedule demands for DWCs such as the opportunity for leave breaks after a period of work, and conducive job schedule times based on individual needs and preferences. In addition, the workplace under which DWCs work must be comfortable enough for them to feel safe and enhance the conditions of work to meet their needs. This flexible work schedule could enable DWCs to meet their basic job requirements and personal responsibilities to help minimize stress related to their job schedules.

## 5. Study Limitations

The study has limitations. Our findings are based on views from DWCs working for two waste companies in the Ho Municipality of Ghana. Hence, the findings may not apply to the wider Ghanaian context. However, the data triangulation afforded by the inclusion of company managers and supervisors’ views could mean that DWCs working for waste management companies in different locations in Ghana may report similar experiences. In addition, given that waste companies ‘organizational culture’ can influence waste management processes, there is possibility that some useful information on the study aim could have been withheld by participants, although research assistants were well trained to minimize such effects. Overall, the process of triangulation of data provided an opportunity to understand and to delve deeper into understanding the aim for the study, thus enriching the findings and minimizing any potential bias in the study.

## 6. Conclusions

DWCs in the study were exposed to health-related risks and reported practicing unsafe behavior regarding occupational safety measures. DWCs were also aware of the enforcement of standard regulation procedures of health hazards associated with solid waste management, however, non-adherence to occupational health and safety rules by waste companies contributed to health-related risks and poor health outcomes. Poor interpersonal relationships among DWCs and their managers and supervisors, and a lack of social protection and job security were other challenges recounted to have negative impacts on DWCs’ psychosocial health needs. The absence of well-negotiated contracts with waste companies and low wages on the job was a contributory factor to work-related stress and poor job satisfaction. Furthermore, the effects of negative attitude, perception, and stigmatization in the community towards DWCs and indiscriminate solid waste disposal lead to job-related stress and poor job satisfaction. There is need for tailored workplace policies that provide psychosocial counseling and support for DWCs to improve how they manage work-related stress. In addition, policy schemes among waste companies that address remuneration challenges, particularly among low-wage workers such as DWCs, can improve the job satisfaction reported in this study. At the national level, policies to ensure the full compliance of waste companies to health worker safety standards, particularly the use of protective clothing and other essential tools to reduce the drudgery of waste collection processes can reduce workload-related stress and improve job satisfaction ratings among DWCs.

## Figures and Tables

**Table 1 ijerph-17-02903-t001:** Demographic and socio-economic characteristics of the study participants.

Characteristics	Company A		Company B	
	Manager/Supervisor	DWC	Manager/Supervisor	DWC
**Age in years**	n	n	n	n
21–30	3	2	2	1
31–40	12	10	4	5
41–50	6	17	2	10
51–60	4	11	2	4
61+	-	3	-	1
**Ethnic Group of participants**				
Ewe	22	42	10	19
Other	3	1	-	2
**Marital Status**				
Single	3	5	3	3
Married	22	27	7	13
Divorced/separated	-	3	-	1
Widow/Widower	-	8	-	4
**Religion**				
Christian	18	27	8	14
Muslim	2	3	-	1
Traditionalist	2	11	1	5
No religion	3	2	1	1
**Educational Level**				
MSLC	-	11	-	5
JSS	-	9	-	5
Vocational training	-	3	-	2
None	-	20	-	9
SHS	15	-	8	-
Tertiary	10	-	2	-
**Gender**				
Female	4	29	2	13
Male	21	14	8	8
**Number of years in the workplace**				
Below 1 year	-	-	-	-
1–5 years	6	8	3	8
6–10 years	17	23	-	13
11–15 years	2	7	-	-
16–20 years	-	5	-	-
21 and above	-	-	-	-

MSLC—Middle School Living Certificate, JSS—Junior Secondary School, SHS—Senior High School, n—number of study participants.

**Table 2 ijerph-17-02903-t002:** Overview of Inductive thematic processes and final themes that emerged among DWC.

Convergent Views from Transcripts	Inductive Step I&II (Open Coding, Cross Validation and Extraction of Significant Statements)	Inductive Step III (Subthemes Selectionand and Verification from Inductions)	Final Themes Emergent
1. Standards and practices of solid waste management.Potential of health risksassociated with the job.Enforcement of healthand safety rules. Preventive measures atworkplace. Knowledge gap at the workbehavior	1.1. Poor adherence to safety standards and practices along the waste collection value chain.1.2. Although most of DWCs acknowledge associated risk with their jobs, knowledge regarding health risk does not translate into practices aimed at reducing any potential risk associated with the job.	Poor adherence to safety standards & practices Risks associated with job, knowledge on potential health risks does not translate into reducing risks	Adherence to and enforcement of safety standards on waste management
2. Relationship among DWCs& supervisors/managers.Job descriptions & schedule.Drivers of health work habits& work output.	2.1. Respectful inter and intra-familial associations among DWCs and managers/supervisors is a positive precursor for driving healthy work habits and outcomes along the waste management value chain. 2.2. Different task roles involve team work and cooperation2.3. To meet this demand for efficiency & reduce any potential work related stress. Follow instructions to reduce any repetitive job duties & work burns.	Relationship among DWCs, managers and supervisors Different tasks involve team work and cooperation in a healthy atmosphere for effective and efficient output to reduce stress	Relationships at work and working conditions
3. Health and Safety at the workplace wellbeing benefits and insurance.	3.1. Psychosocial well-being and overall work balances affect their health.3.2. Displeasure about current arrangements that ignore largely their health needs at the expenses of improving environmental safety & hygiene.3.3. DWCs paid for their own health insurance, neither were they guaranteed of any medical refund as a results of any ill-health resulting from work.	Inadequate care of health and wellbeing at the expense of keeping the environment clean,None benefits of health insurance and difficulties in getting medical bills refund	Social protection and job insecurity
4. Job satisfaction /dissatisfaction.Motivation, psychosocial risks, and work-relatedstress among DWCs.Capacity building and training.	4.1. Negative attitude and perception of the community towards DWCs, families and friends. Poor understanding of the burdensome of the job often results in uncontrolled stress and poor job satisfaction for DWCs.4.2. DWCs perceive that community members view them negatively and they see DWCs as people working with refuse, are people that do not have integrity, and call them all types of names(‘Borla Borla’) as well as making derogatory comments about them.	Poor job dissatisfaction & motivation Negative attitudes and perceptions of community towards DWCs, families and friends Poor understanding of the job, and DWCs being treated differently, capacity building, psychosocial and economical stress	Job satisfaction

## Data Availability

The data generated, analyzed and presented in this study are available from the authors upon request.

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
