# Peer review of "Psychosocial Risk, Work-Related Stress, and Job Satisfaction among Domestic Waste Collectors in the Ho Municipality of Ghana: A Phenomenological Study"

_ijerph, 2020, doi:10.3390/ijerph17082903_

Round 1
Reviewer 1 Report
The objective of the study is not stated, I identify that it could be To contribute further to the understanding of the occupational needs and hazards among DWCs the context of solid waste collection and disposal in Ghana or more specifically to understand DWCs 'and their managers and supervisors' views and experiences regarding risk, work-related stress, and job satisfaction needs along the waste management value chain in the study setting. The study presents various conclusions related to the 4 topics investigated I) Adherence to and enforcement of safety standards on waste management; II) Relationships at work and working conditions; III) Social protection and job insecurity, and IV) Job satisfaction identifying various areas of opportunity to improve the health and well-being of DMCs
On the other hand, the study design is pointed out from the title, however, some aspects are not clear.
1. In paragraph 119 it is mentioned that 70 DWC met the eligibility criteria, however, these criteria are not described, so it is not known if these 70 DWC are the totality of the DWC of both companies. Only inclusion criteria for supervisors are specifically mentioned.
2. In the Study instrument section requires more information about the instrument used, what do you call minimum corrections on line 112-113? Which version of the Copenhagen Psychosocial Questionnaire (COPSOQ) scales was used: Long, Medium-length or Short questionnaire? More information is also needed about the Focus Groups Discussion. How many people participated in each group, what was the dynamics of participation?
3. In the Sampling and interview procedures section, what does it mean that 3 research assistants were selected based on their level of education or prior experience in conducting research? Was there an assessment of interobserver variability?
With respect to the results and the discussion, the opportunity to identify differences in the four identified themes in which the demographic variables registered could affect, such as sex and age, is lost.
Table 1 does not have a table footer that explains the abbreviations used, it should be noted that the numbers presented are n and not proportions.
Author Response
Reviewer 1
Reviewer 1 Comment 1
‘’The objective of the study is not stated, I identify that it could be To contribute further to the understanding of the occupational needs and hazards among DWCs the context of solid waste collection and disposal in Ghana or more specifically. The study presents various conclusions related to the 4 topics investigated I) Adherence to and enforcement of safety standards on waste management; II) Relationships at work and working conditions; III) Social protection and job insecurity, and IV) Job satisfaction identifying various areas of opportunity to improve the health and well-being of DMCs. On the other hand, the study design is pointed out from the title; however, some aspects are not clear.
1. In paragraph 119 it is mentioned that 70 DWC met the eligibility criteria, however, these criteria are not described, so it is not known if these 70 DWCs are the totality of the DWC of both companies. Only inclusion criteria for supervisors are specifically mentioned”
Author’s Response to comment 1
Based on the above comment, a new paragraph has been introduced as found in line 81-83 on page 2. The new paragraph reads “Specifically, the objective of the study was to understand the views and experiences of DWCs and their managers’or supervisors regarding personal health and safety risks, work-related stress, and job satisfaction needs in two waste companies in Ghana.”
Paragraph 119 referred here by reviewer 1 regarding the eligibility criteria of DWCs appears now in line 142-145 on page 3 and page 4 of the revised manuscript. The text has been revised in line 142-145 to now reads ”Overall, 70 DWCs met the eligibility criteria of having minimal work experience as DWC of one-year. Different work profiles among the DWCs were included in the study such as cleaners, sweepers, janitors, and drivers who worked for the two waste companies handling solid waste along the value chain, 46 from Company A and 24 in Company B.”
Reviewer 1 Comment 2
“In the Study instrument section requires more information about the instrument used, what do you call minimum corrections on line 112-113?
“Which version of the Copenhagen Psychosocial Questionnaire (COPSOQ) scales were used: Long, Medium-length or Short questionnaire”?
“More information is also needed about the Focus Groups Discussion. How many people participated in each group, what was the dynamics of participation”?
Authors Response to comment 2
The minor correction in line 112-113 refers to a typo error that was found in the instrument during the pilot testing. This was corrected in the final instrument for data collection. In the revised manuscript, what constituted minor corrections have been explained in brief in lines 136 and now reads “No content changes were made to the instrument except for some small textual edits.”
Further details of the Copenhagen Psychosocial Questionnaire (COPSOQ) scales applied in this study is provided in lines 129-132 in this revised submission and now reads; “The study adapted the short, medium and the long versions of the Copenhagen Psychosocial Questionnaire (COPSOQ) scales. All the versions were modified to meet the objective(s) of this study. Specifically, we created a mix of scales to measure psychosocial risk, work-related stress and job satisfaction.”
On how participants were involved during FGDs, the following revisions have been made and is reflected in the revised manuscript in lines 173-177 The revised version now read as; ‘’Six focus group discussions were conducted as follows: four FGDs was conducted in company A while two FGDs were conducted in company B. Four of the six FGDs had ten participants each participating in the discussions. Two FGDs, one in company A had 13 participants while another FGD in company B consisted of 11 participants. The use of focus discussion made the participants express their opinions and real-life experiences on research topics
Reviewer 1 Comment 3
“In the Sampling and interview procedures section, what does it mean that 3 research assistants were selected based on their level of education or prior experience in conducting research? Was there an assessment of interobserver variability”?
“With respect to the results and the discussion, the opportunity to identify differences in the four identified themes in which the demographic variables registered could affect, such as sex and age, is lost”.
“Table 1 does not have a table footer that explains the abbreviations used, it should be noted that the numbers presented are n and not proportions”
Authors Response to comment 3
Regarding clarity on what determined the selection of RAs and their roles, the text has been revised in the new submission in lines 154-165 and now reads; “Three research assistants (RAs) were recruited to assist in data collection based on their level of education, prior demonstrated experience and performance during training in conducting field research observations and a good understanding of ethical procedures in research. Fluency in the local language (Ewi) and good background knowledge of the study setting was also considered in recruiting RAs. The RAs were trained on how to undertake field observations and to administer informed consent to study participants. The principle investigator led all interviews while RAs assisted to take short notes during the conduct of these interviews. Short notes from the three RAs were later corroborated with audio recordings after transcriptions were concluded and there were no significant differences between the transcripts and the notes from all RAs. During field data collection, RAs visited dumping sites and documented in a notebook field observation on types of solid waste generated, methods of solid waste collection and disposal, as well as the use of Personal Protective Equipment (PPE)”
On the interobserver variability comment, RAs roles in the study involved administering consent forms and follow-up observations to dumping sites. These roles were minimal and did not have any potential effects on introducing variability in relation to the findings from this study. To provide clarity on the comment raised by the reviewer, a new sentence has been incorporated in lines 180-184 and now reads as;
“The principal investigator led all interviews while RAs assisted to take short notes during the conduct of interviews and field observations. Short notes from the three RAs were later corroborated with audio recordings after transcriptions were concluded and there were no significant differences between the transcripts and the notes from all RAs”
In the results section, from lines 164-172 the proportion to participants based on gender and age differentials was provided. All analysis conducted was for the entire groups, that is to say, either DWCs, or managers/supervisors without any segregation on the basis of the sex, age, etc. Given that this study was qualitative in nature, its design was not to measure its objective in relation to demographic characteristics such as age, sex, etc. This explains why the results and discussion was not done based on sex and age variations. This is a good and fair comment and would be taken into consideration during future quantitative studies. Hence, no revision was made with respect to this comment on the revised submission. However, in the discussion section, further details of gender differentials are provided in lines 439-452 on page 11.
In Table 1, a footer has been provided and abbreviations defined for clarity on lines 214-215. The following has been inserted at the footer; “MSLC-Middle School Living Certificate, JSS-Junior Secondary School, SHS-Senior High School, n-number of study participants”
Reviewer 2 Report
Thank you for the opportunity to review the manuscript “Psychosocial risk, work-related stress, and job satisfaction among domestic waste collectors in the Ho Municipality of Ghana: A phenomenological study”.
Congratulations to the authors for their work, I found your paper a potentially very valuable resource on Health Science and therefore an interesting and relevant contribution to IJERPH.
The manuscript investigated psychosocial risk factors, work-related stress and job satisfaction needs among municipal solid waste collectors in the Ho Municipality of Ghana.
However, in my opinion there are several aspects should be revised to improve the explanatory power of the manuscript as noted below.
SPECIFIC COMMENTS:
TITTLE
Correct.
ABSTRACT
In the abstract the results and conclusions seem mixed. It is recommended to structure the information.
INTRODUCTION
Correct.
METHOD AND RESULTS
To maintain the anonymity of the informants, they should be coded and not report age on verbatim
DISCUSSION AND CONCLUSION
Correct
Author Response
Reviewer 2
Comments from Reviewer 2
Congratulations to the authors for their work, I found your paper a potentially very valuable resource on Health Science and therefore an interesting and relevant contribution to IJERPH. The manuscript investigated psychosocial risk factors, work-related stress and job satisfaction needs among municipal solid waste collectors in the Ho Municipality of Ghana.
However, in my opinion, there are several aspects should be revised to improve the explanatory power of the manuscript as noted below.
ABSTRACT: In the abstract, the results and conclusions seem mixed. It is recommended to structure the information.
METHOD AND RESULTS: To maintain the anonymity of the informants, they should be coded and not report age on verbatim.
Author’s Response to Reviewer 2
Thank you for acknowledging the importance of this piece of work sent for review and for providing us some useful comments to improve the manuscript.
In the abstract section, few text revisions have been made to bring clarity on the results and the conclusions of the study. In line, 24 a clear statement has been inserted (The study results) to provide clarity on the results from this study. Also, in line 32 a clear statement (In conclusion, the findings) has also been introduced to provide more structure to the abstract.
Anonymity of the informants was assured as described in line 99-106: Participants’ identities were concealed by ensuring that participants do not mention their names on tape during the recording and interviewing processes. The study participants were assured that the information provided is handled with confidentiality and analysis of the data is done at the aggregate level of the group to ensure anonymity. Also, the research team assured the participants that the information shared with the researchers will not be used to harm their job in the future. This generated a lot of confidence among the study participants and ensured a high level of participation during the interviews.
Reviewer 3 Report
This manuscript presents an original study about psychosocial risk factors, work-related stress and job satisfaction needs among municipal solid waste collectors in the Ho Municipality of Ghana. Is well structured, using a clear style, and justify properly the methodological choices made. The review of research is good, the results section is clear and well-structured and the discussion, despite not bringing any significant novelty in the field of public health, is solid and well-founded.
The main limitation, assumed by the authors, is the fact that findings may not apply to the wider Ghanaian context. However, the data triangulation afforded by the inclusion of company managers and supervisors’ views have contributed to reducing this risk, mean that major outcomes of similar research in different locations in Ghana may report similar experiences.
In order to improve the final result, some aspects concerning the following sections should be changed / clarified:
Introduction:
- Add and explore the concept of «organizational culture», that is used for discussion and necessary for understanding the study main objective;
Materials and methods:
- Justify the choice of Ho Municipality to do this research;
- Line 23: «Data were 22 analyzed using inductive and deductive content procedures to form themes based on the study aim»
It will be useful to clarify what was the expression of the deductive dimension in the study, once we are facing perspectives and lived experiences to understand the phenomenon.
- Line 83: Ethical approval - Because this study covers a particularly sensitive of labor relationships, with asymmetric hierarchies and antagonistic interests (companies, managers / supervisors and workers), ethics issues and how they are adressed in the research process should be clearly stated in the manuscript.
- Line 113: authors wrote «Interviews were conducted until data saturation reached»; and in line 119 «Overall, 70 DWCs met the eligibility criteria, 46 from Company A and 24 in Company B. However, six (6) DWCs were excluded because of their non-availability at the time of data collection. We had 64 DWCs for the interviews».
It is necessary to clarify how many interviews authors did to reach the saturation point or whether all 64 were interviewed. That is, clarify which criterion was followed in the process of conducting the interviews (saturation point or previously defined number).
- Lines 160/163: «Most of DWCs were females (66%)». «Besides DWCs, a total of 23 supervisors 163 and 12 managers in the two companies took part in the study. Approximately 82.9% of the managers 164 and supervisors were males».
In view of the gender inequality in the occupation of hierarchical positions, it would be interesting to have a comment that explores this difference.
Results:
- Line 259: I suggest a title change . The title «Non-clarity on job roles and conflicting job demands» of this subpoint does not match the content, more related to continuing training.
- The paragraph starting at line 332 and the paragraph starting at line 370 contain contradictory statements, apparently. It would be important for the authors to further clarify these statements.
Discussion:
- Line 487: the concept of flexible schedule work is referred to but not defined or explained. Given its importance, a more developed presentation of this concept would be useful.
Author Response
Reviewer 3
“This manuscript presents an original study about psychosocial risk factors, work-related stress and job satisfaction needs among municipal solid waste collectors in the Ho Municipality of Ghana. Is well structured, using a clear style, and justify properly the methodological choices made. The review of research is good, the results section is clear and well-structured and the discussion, despite not bringing any significant novelty in the field of public health, is solid and well-founded. The main limitation, assumed by the authors, is the fact that findings may not apply to the wider Ghanaian context. However, the data triangulation afforded by the inclusion of company managers and supervisors’ views have contributed to reducing this risk, mean that major outcomes of similar research in different locations in Ghana may report similar experiences. In order to improve the final result, some aspects concerning the following sections should be changed / clarified:”
Reviewer 3 Comment 1
Introduction and method sections
Add and explore the concept of «organizational culture» that is used for discussion and necessary for understanding the study main objective;
Author’s Response
In the introduction section, the word organizational culture has been introduced and incorporated in lines 51-53 that now reads; “In the PPP arrangement, an organizational culture was built in which the public actors such as MMDAs work with large to medium size private companies who employ Domestic Waste Collectors (DWCs) to collect and dispose of solid waste”
Also, in lines 54-57, some text revisions to include organizational culture has been done and now reads as; “Evidence in Ghana suggests poor organizational culture attributes such as poor stakeholder engagements, lack of transparency in waste management contracts, and poor quality assessment procedures regarding solid waste management in Ghana [10–12].”
Other sections of the introduction, lines 64, 65 have all highlighted organizational culture issues and its relations to the aim/objective of this study.
Reviewer 3 Comment 2
Materials and methods:
- Justify the choice of Ho Municipality to do this research;
Author’s Response
The choice of Ho Municipality as the study setting is provided in lines 112-116 as presented here; “The Municipality was chosen for the study because of its solid waste management challenges and the state of environmental conditions in the municipality and a lot more needs to be done”
Reviewer 3 Comment 3
“- Line 23: «Data were 22 analyzed using inductive and deductive content procedures to form themes based on the study aim» It will be useful to clarify what was the expression of the deductive dimension in the study, once we are facing perspectives and lived experiences to understand the phenomenon”.
Author’s Response
To provide clarity on the deductive processes leading to the emergence of final themes in the study, the manuscript has been revised and Table 2 presents further details on how each of the final themes emerged. Table 2 has been added separately and should be inserted on in line 237 on page 7 of the manuscript on page.
Reviewer 3 Comment 4
- Line 83: Ethical approval - Because this study covers a particularly sensitive of labor relationships, with asymmetric hierarchies and antagonistic interests (companies, managers / supervisors and workers), ethics issues and how they are addressed in the research process should be clearly stated in the manuscript.
Author’s Response
In the methods section, under study design and ethical approval, further details have been enumerated on the processes that were undertaken to ensure participant confidentiality and to guarantee the anonymity of study respondents. The revised section from lines 95-106 now reads as;
“The Ethical Review Committee Psychology and Neuroscience (ERCPN) of the Faculty of Psychology and Neuroscience, Maastricht University (approved research line ERCPN-188_10_02_2018) and Ghana Health Service Ethics Review Committee (GHSERC 08/05/17) approved the study. During the study, each participant completed a consent form before participating, after being informed about the objective and procedure of the study. Participants’ identities were concealed by ensuring that participants do not mention their names on tape during the recording and interviewing processes. The study participants were assured that the information provided will be handled with confidentiality and analysis of the data will be done at aggregate level to ensure anonymity. Also, the research team assured the participants that the information shared with the researchers will not be used to harm their job in the future. This generated a lot of confidence among the study participants and ensured a higher level of participation during the study”
Reviewer 3 Comment 5
- Line 113: authors wrote «Interviews were conducted until data saturation reached»; and in line 119 «Overall, 70 DWCs met the eligibility criteria, 46 from Company A and 24 in Company B. However, six (6) DWCs were excluded because of their non-availability at the time of data collection. We had 64 DWCs for the interviews». It is necessary to clarify how many interviews authors did to reach the saturation point or whether all 64 were interviewed. That is, clarify which criterion was followed in the process of conducting the interviews (saturation point or previously defined number).
Authors’ Response
In lines 113 and line 119 of page 3 of the submission clarity has been provided on how many interviews were conducted across the two waste management companies in the study. The sentences also provide the number of participants in each of those interviews. In conducting the interviews, the defined number (6 interviews) was applied. This is reflected in lines 173-176 of page 4 as presented here;
‘’Six focus group discussions were conducted as follows: four FGDs were conducted in company A while two FGDs were conducted in company B. Four of the six FGDs had ten participants each participating in the discussions. Two FGDs, one in company A had 13 participants while another FGD in company B consisted of 11 participants.
Regarding the matter of saturation, the sentence in lines 177-180 to addresses the review concern on saturation of the interviews;
“All six FGDs were conducted at separate times and locations. In addition, 35 IDIs were conducted among managers and supervisors. For each IDI respondent, interviews were conducted until at a point where saturation was reached among each interviewee.”
Reviewer 3 Comment 6
- Lines 160/163: «Most of DWCs were females (66%)». «Besides DWCs, a total of 23 supervisors 163 and 12 managers in the two companies took part in the study. Approximately 82.9% of the managers 164 and supervisors were males».
In view of the gender inequality in the occupation of hierarchical positions, it would be interesting to have a comment that explores this difference.
Author’s Response
In the discussion section, further details of gender differentials is provided in lines 487-500 of page 14.
“Although the study was not aimed to assess gender differentials, we highlight some key gender issues that are worth mentioning. Fewer females constituted the study participants at the level of managers and supervisors compared to males. This is directly the opposite regarding DWCs since more females constituted this level of study participants. DWCs job roles are the lowest along the waste management value chain, indicating that females tend to occupy the lower job roles in managing waste. This finding is consistent with current waste management operations across Ghana. Some reasons such as culture, socio-cultural value systems that regard jobs such as cleaning, sweeping and waste collection as traditional roles for women may be influencing low socio-economic women desire for domestic waste collection job roles as seen from this study. Policies that will empower women working in the waste management sector to take a leadership position and assume higher responsibility and roles will support to enhance family livelihoods, decision making and bridge the inequality job roles specifications in the waste management sector in Ghana and across similar settings”
Reviewer 3 Comment 6
Results:
- Line 259: I suggest a title change. The title «Non-clarity on job roles and conflicting job demands» of this sub point does not match the content, more related to continuing training.
The paragraph starting at line 332 and the paragraph starting at line 370 contain contradictory statements, apparently. It would be important for the authors to further clarify these statements.
Author’s Response
The sub-title has been revised and now reads; ‘’ Non-clarity on job roles and in-service training’’ that can be found in line 361.
In lines, 474-476 text changes have been made to address the concern raised. The new revision reads as ; “DWCs recounted that some community members were happy of their valuable service in their communities despite work-related hazards and poor remuneration issues affecting them as DWCs”
Reviewer 3 Comment 7
Discussion:
- Line 487: the concept of flexible schedule work is referred to but not defined or explained. Given its importance, a more developed presentation of this concept would be useful.
Author’s Response
In revising the manuscript, the following sentence has been incorporated in line 612-614 of page 16 to define, what flexible scheduled work is talked about earlier in line 487.
The revised section that defines flexible scheduled work now reads as; “There is need for workplace policies that address flexible work schedule demands for DWCs such as opportunity for leave breaks after period of work, and conducive job schedule times based on individual needs and preferences”